# Development of a new device for manipulating frozen mouse 2-cell embryos on the International Space Station

Sayaka Wakayama[1,2]*, Mariko Soejima[2], Yasuyuki Kikuchi[2], Erika Hayashi[2], Natsuki Ushigome[2], Ayumi Hasegawa[3], Keiji Mochida[3], Tomomi Suzuki[4], Chiaki Yamazaki[4], Toru Shimazu[5], Hiromi Sano[6], Masumi Umehara[7], Hitomi Matsunari[8,9], Atsuo Ogura[3], Hiroshi Nagashima[8,9], Teruhiko Wakayama[1,2]*

1 Advanced Biotechnology Center, University of Yamanashi, Yamanashi, Japan, 2 Faculty of Life and Environmental Sciences, University of Yamanashi, Yamanashi, Japan, 3 RIKEN BioResource Research Center, Tsukuba, Ibaraki, Japan, 4 Japan Aerospace Exploration Agency, Tsukuba, Japan, 5 Space Utilization Promotion Department, Japan Space Forum, Tokyo, Japan, 6 Japan Manned Space Systems Corporation, Tokyo, Japan, 7 Advanced Engineering Services Co., Ltd, Tsukuba, Ibaraki, Japan, 8 Laboratory of Developmental Engineering, Department of Life Sciences, School of Agriculture, Meiji University, Kawasaki, Japan, 9 Meiji University International Institute for Bio-Resource Research (MUIIBR), Kawasaki, Japan

* sayakaw@yamanashi.ac.jp (SW); twakayama@yamanashi.ac.jp (TW)

**Data Availability Statement:** All relevant data are within the paper.

## Abstract

Whether mammalian embryos develop normally under microgravity remains to be determined. However, embryos are too small to be handled by inexperienced astronauts who orbit Earth on the International Space Station (ISS). Here we describe the development of a new device that allows astronauts to thaw and culture frozen mouse 2-cell embryos on the ISS without directly contacting the embryos. First, we developed several new devices using a hollow fiber tube that allows thawing embryo without practice and observations of embryonic development. The recovery rate of embryos was over 90%, and its developmental rate to the blastocyst were over 80%. However, the general vitrification method requires liquid nitrogen, which is not available on the ISS. Therefore, we developed another new device, Embryo Thawing and Culturing unit (ETC) employing a high osmolarity vitrification method, which preserves frozen embryos at −80°C for several months. Embryos flushed out of the ETC during thawing and washing were protected using a mesh sheet. Although the recovery rate of embryos after thawing were not high (24%-78%) and embryonic development in ETC could not be observed, thawed embryos formed blastocysts after 4 days of culture (29%-100%) without direct contact. Thus, this ETC could be used for untrained astronauts to thaw and culture frozen embryos on the ISS. In addition, this ETC will be an important advance in fields such as clinical infertility and animal biotechnology when recovery rate of embryos were improved nearly 100%.

**Funding:** This work was partially funded by the Naito Foundation (S.W) and Takahashi Industrial and Economic Research Foundation (189) to S.W.; Asada Science Foundation (T.W) and the Canon Foundation (M20-0006) to T.W. The authors would like to thank Enago for the English language review. The funders had no role in study design, data collection and analysis, decision to publish, or preparation of the manuscript.

**Competing interests:** The authors have declared that no competing interests exist.

## Introduction

Sustaining life beyond Earth on space stations or on other planets will require a comprehensive understanding of how the characteristics of the environment in space, such as microgravity (μG) or space radiation, affect key phases of mammalian reproduction [1]. Performing experiments on the reproduction of mammals is difficult, and most studies on reproduction in space are therefore limited to nonmammalian species such as fish or amphibians [2–8]. However, the reproductive systems of the latter are fundamentally different from those of mammalian viviparity, which comprises pre- and post-implantation phases of embryonic development accompanies by the formation of the placenta. Therefore, the research outcomes of such studies cannot be extrapolated to advance our understanding of mammalian reproduction in space.

Previous studies conducted on the International Space Station (ISS) compared with those on Earth show that ionizing radiation in space will not affect for the quality of mammalian spermatozoa and their offspring for more than 200 years when they are preserved using freeze-drying [9, 10]. However, it is not possible to generate μG for long periods on Earth, and the effects of μG on mammalian reproduction are therefore unknown. We previously considered the possibility of conducting research on mammalian embryonic development *in vitro* under simulated μG using a three-dimensional clinostat [11]. Embryo culture for 4 days under such conditions generates significant anomalies such as delayed embryonic development, deterioration of the trophectoderm, impaired rates of differentiation, and failure to develop viable offspring [12]. Similar results were published by other groups [13, 14], suggesting that μG may adversely influence the development of preimplantation mammalian embryos. Thus, humans may not succeed in permanently colonizing other planets because of failed reproduction.

The abovementioned experiments were simulations performed on the ground. We aimed to determine whether the same result would be obtained in the μG of real space, such as the ISS. Frozen embryos must be launched, thawed, and cultured on the ISS to study the effects of μG on embryonic development, because mouse embryos can be cultured *in vitro* for only 4 to 5 days. However, two factors hindered the application of such experiments on the ISS. First, handling of embryos by untrained personnel not only in orbit but also on the ground is not feasible given their small size (80–100 μm). If the cells are sticking to the bottom of the culture device, they will not flow out when culture medium was exchanged using a pump or syringe by astronauts on the ISS [15]. However, considering that the embryos are floating in the medium, they can easily flow out when medium was changed. Therefore, developing a new device that prevents embryos from flowing out when astronauts perform space experiment is necessary.

Another factor hindering the application of space experiments on the ISS is the cryopreservation temperature of frozen embryos. Frozen embryos should be maintained below −130˚C during cryopreservation to avoid cryodamage [16, 17]. If embryos are exposed to high temperature, intracellular ice crystals form, thereby irreversibly damaging the embryos. However, liquid nitrogen ($LN_2$) is not available on the ISS, and the temperature of the coldest freezer is −95˚C. Furthermore, the frozen embryos must be preserved for a few weeks or months on the ISS before the commencement of experiment, because space experiments cannot be performed immediately after arrival at the ISS. During the study, Mochida et al. developed high-osmolality vitrification (HOV), which enable the storage of the embryos at −80˚C for several months [18].

In this study, we developed a simple system for thawing and culturing embryos to address the aforementioned challenges, which will allow us to preserve embryos at −80˚C for a few

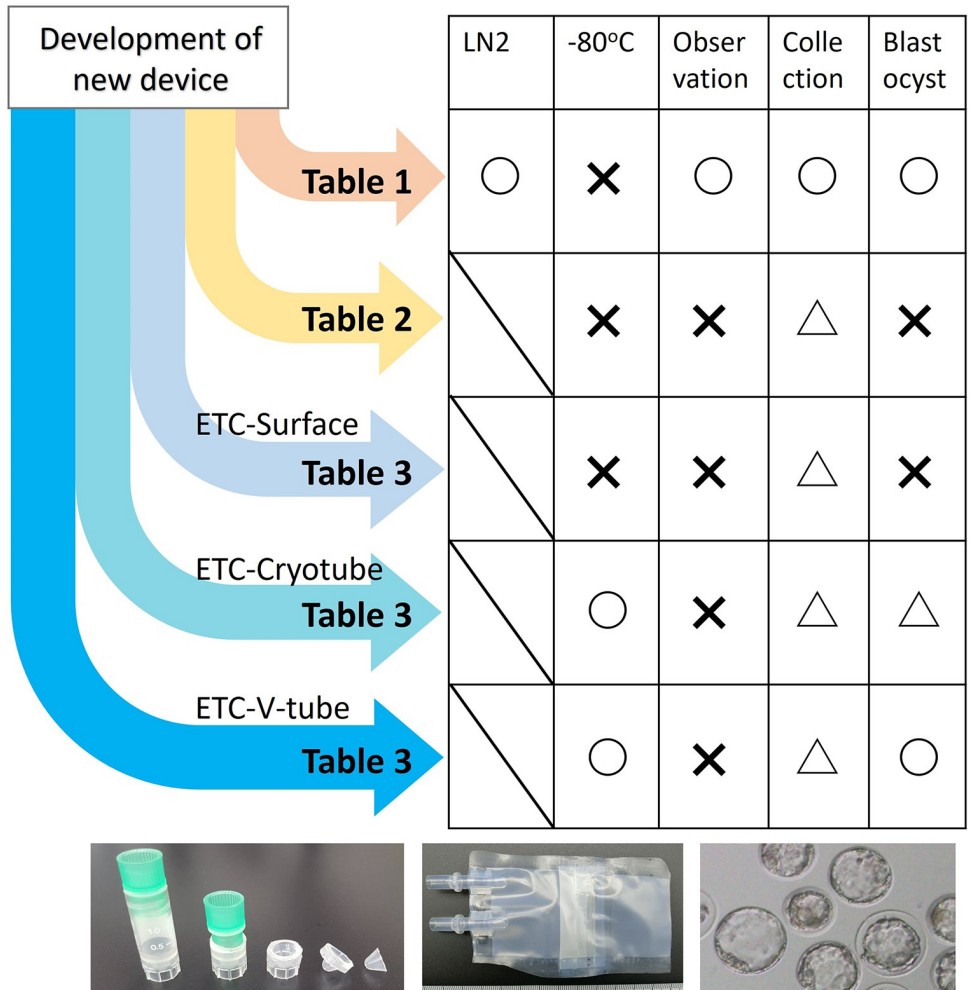

| | LN2 | -80°C | Observation | Collection | Blastocyst |
|---|---|---|---|---|---|
| **Table 1** | ○ | ✕ | ○ | ○ | ○ |
| **Table 2** | | ✕ | ✕ | △ | ✕ |
| ETC-Surface **Table 3** | | ✕ | ✕ | △ | ✕ |
| ETC-Cryotube **Table 3** | | ○ | ✕ | △ | △ |
| ETC-V-tube **Table 3** | | ○ | ✕ | △ | ○ |

**Fig 1. Study outline.** Diagram showing the performance and weaknesses of the device, completed devices, and blastocysts.

months and untrained personnel, such as astronauts, to perform embryo experiments on Earth and on the ISS (Fig 1).

## Results

### OptiCell device

In the first attempt, we used a hollow fiber tube (HFT) [19] and an OptiCell device. Mouse 2-cell embryos were inserted into the HFT, which was then inserted into a pipette tip attached to a 1-ml syringe and then immersed in $LN_2$ (Fig 2a–2d). The frozen embryos in the HFT were inserted into the OptiCell immediately after thawing (Fig 2e and 2f) and then washed and cultured through exchange with the rewarming solution and culture medium respectively. Those solutions were delivered using a 10-ml syringe (Fig 2g) and then placed in an incubator (37°C, 5% $CO_2$) for 4 days. The HFT in the OptiCell achieved 88% survival of thawed embryos (Table 1: OptiCell).

Blastocytes developed from 82% of the 2-cell embryos after 4 days in culture, which closer to non-frozen control embryos (92%). Generally, thawing frozen embryos is strictly time-

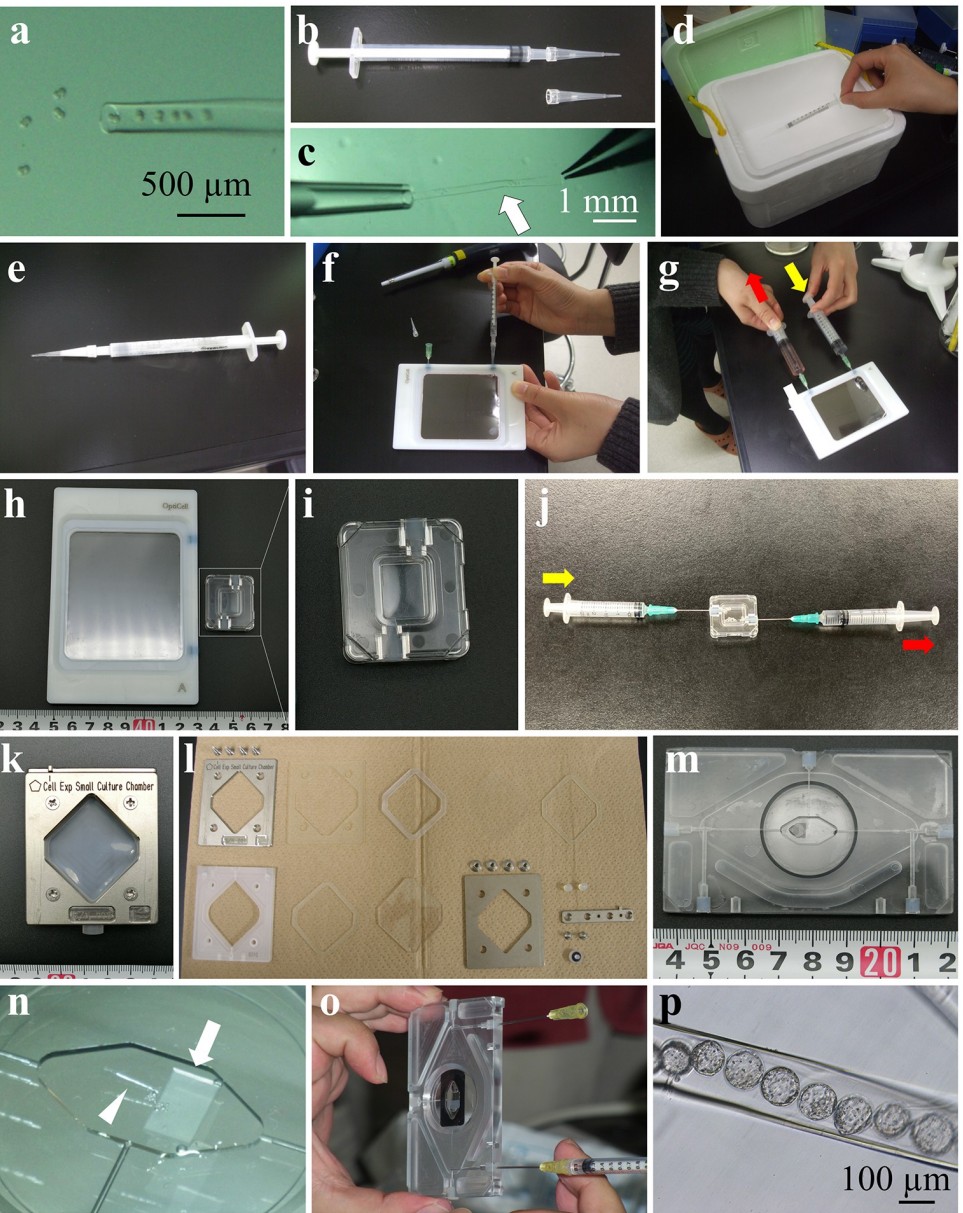

**Fig 2. Vitrification devices with hollow fiber tube.** (a) Mouse 2-cell embryos were aspirated into the HFT. (b) Pipette tip attached with syringe. (c) An HFT with embryos was inserted into a pipette tip. Arrow indicates the HFT with embryos. (d) The syringe was immersed into $LN_2$. (e) Immediately after removal from $LN_2$. (f) The HFT inside the pipette tip was inserted into the port of an OptiCell. (g) Solution/medium exchange using a 10-ml syringe via the port. (h) OptiCell (left) and Mini-plate (right). (i) High magnification of the Mini-Plate. (j) Solution/medium exchange via the port. (k) Cell Experimental Unit (CEU). (l) Disassembled CEU. (m) New Cell Experimental Unit (NCEU). (n) The NCEU was equipped with a silicone flipper (arrow) used to pinch the HFT (arrowhead) onto the center of the plate. (o) Solution/medium exchange. (p) Blastocysts inside the HFT. Embryos were harvested from the NCEU after 4 days.

limited and requires intensive practice. However, this method could be performed without any practice, therefore it will be use in animal research facilities and infertility clinics. However, if the HFT was located on the inside edge of the OptiCell, the embryos were difficult observe. Although hard shaking moved the HFT from edge to the center of the OptiCell, this

**Table 1. Recovery and blastocyst rates of embryos frozen and thawed in new devices using a hollow fiber tube.**

| Device | Freezing | No. of embryos Vitrified | No. (%) of embryos retrieved | No. (%) of embryos survived | 4–8 cell (%) | Morula (%) | Blastocyst (%) |
|---|---|---|---|---|---|---|---|
| Culture Dish | No | - | - | 86 | 86 (100) | 85 (99) | 85 (99)a |
| | Yes | 111 | 110 (99) | 91 (83) | 71 (78) | 63 (69) | 49 (54)b |
| OptiCell | No | - | - | 61 | 61 (100) | 61 (100) | 56 (92) |
| | Yes | 61 | 57 (93) | 50 (88) | 50 (100) | 48 (96) | 41 (82) |
| Mini plate | No | - | - | 52 | 52 (100) | 51 (98) | 51 (98) |
| | Yes | Failed | - | - | - | - | - |
| CEU | No | | | 59 | 57 (97) | 57 (97) | 56 (95) |
| | Yes | Failed | - | - | - | - | - |
| NCEU | No | | | 50 | 50 (100) | 50 (100) | 50 (100) |

Numbers with different superscripts (a, b) within the same column are significantly different (P < 0.01).

CEU: Cell Experimental Unit.

NCEU: New Cell Experimental unit.

was difficult. Furthermore, the culture area of the OptiCell (100 cm$^2$) was too large, making it difficult to find the HFT inside the OptiCell using a microscope. The astronauts performing this experiment on the ISS must be able to easily to find the HFT inside the device, which requires limiting the culture area.

## Mini-plate device

We next used a Mini-plate developed by the Japan Aerospace Exploration Agency (JAXA) instead of the OptiCell. The structure of the Mini-plate is similar to that of the OptiCell, but much smaller (Fig 2h and 2i). When fresh embryos in the HFT were inserted into the Mini-plate as a control and cultured for 4 days in the $CO_2$ incubator, 98% of the embryos developed into blastocysts, demonstrating that the Mini-plate was not cytotoxic (Table 1: Mini plate). However, the HFT more often resided on the edge of this device and was difficult to move to the center, even after vigorous shaking. Furthermore, the port of Mini-plate is too tight, requiring the use of a metal needle instead of pipette tip to insert the HFT (Fig 2j). However, when the HFT was frozen inside the metal needle, the HFT cracked after thawing, and all embryos were lost during washing. To avoid cracking the HFT in the needle, the HFT was inserted into the Mini-plate before immersion in the vitrification solution. However, we were unable to perform vitrification within the 2-min limit, and none of the thawed embryos survived.

## Cell Experimental Unit (CEU)

We next used a Cell Experimental Unit (CEU) developed by JAXA in place of the Mini-plate (Fig 2k). The CEU can be assembled so that the HFT harboring embryos can be fixed to its center. When non-frozen embryos in the HFT were cultured in the CEU, 95% developed into blastocysts after 4 days of culture in the $CO_2$ incubator (Table 1: CEU), indicating that the CEU was not toxic to the embryos. However, because of the complex structure of the CEU (Fig 2l), the HFT containing frozen embryos melted before CEU assembly was completed, even on dry ice. We attempted to place the HFT containing embryos into the vitrification solution onto the plate of CEU before freezing and then assembly, after which the CEU was immersed in $LN_2$. Unfortunately, we were unable to complete the process within the time limit, despite practice. We therefore abandoned the use of the CEU.

## New Cell Experimental Unit (NCEU)

Next, we designed a new device based on the CEU, which we called the NCEU (Fig 2m). The NCEU, which tolerates freezing in $LN_2$, is equipped with a silicone flipper that can pinch the HFT onto the center of plate (Fig 2n). The culture area of the NCEU was easily covered by a gas-permeable membrane, which was attached using a nontoxic adhesive. When the HFT with fresh embryos was pinched onto the center of the NCEU and the medium was changed three times (Fig 2o), the HFT could not be flushed out, and 100% of the embryos developed to blastocysts (Fig 2p, Table 1: NCEU) after 4 days of culture in the $CO_2$ incubator. When OptiCell method was used, sometimes HFT was located on the inside edge and became difficult to observe embryos. However, this method allows the HFT to be fixed in the center and at the bottom of plate, making it easy to observe all embryos. Thus, we concluded that the NCEU was possibly the best device for thawing, washing, and observing embryos and that the procedures could be successfully performed without practice.

## Restriction of freezer on ISS

Even if the frozen embryos are transported by rocket to the ISS, frozen embryos must be stored on the ISS for about a month before start experiment because the astronauts are too busy immediately after the rocket's arrival. During the development of these devices, we were informed that liquid nitrogen was not available on the ISS. When using general vitrification method, the -95°C freezer at ISS cannot preserve frozen embryos, even for short periods of time. Therefore, we decided to use the high osmolarity vitrification (HOV) method, which allows storage of the embryos at −80°C for >1 month [18]. However, because the pore size of the membrane of the HFT was very small (5–15 nm), the vitrification solution (EFS42.5) used in the HOV method is highly viscous and cannot be exchanged with the thawing solution (0.75 M sucrose-PB1) through the HFT membrane within time limit. Therefore, we decided to start developing a new method that does not use HFT.

## Frozebag using a mesh bag

To achieve exchange of the viscous solution without flushing out the embryos, we used a mesh sheet (Fig 3a), which was converted into a bag using a heat sealer (hereafter "mesh bag") (Fig 3b). A Frozebag (Fig 3c) served as the embryo culture device. First, to exam toxicity, we directly inserted fresh embryos into the Frozebag, filled it with culture (CZB) medium, and the heat-sealed it. However, none of the embryos developed to the blastocysts, likely because of the bag's impermeability to gas (Table 2). Therefore, we developed a new embryo culture system, in which CZB medium was equilibrated with 5% $CO_2$ in a $CO_2$ incubator before use (hereinafter referred to as eCZB medium) (Fig 3d) [20]. Next, fresh embryos were directly placed in the Frozebag, or inserted into the mesh bag on the scoop (Fig 3e), placed in the Frozebag, and then cultured for 4 days in eCZB medium (Fig 3f). Under these conditions, 86–94% of embryos developed into blastocysts (Table 2), which demonstrated that the Frozebag, the mesh bag, and the scoop were not cytotoxic.

Next, the mesh bag on the scoop was filled with EFS42.5, and 2-cell embryos were inserted into the mesh bag, frozen in $LN_2,$ and the scoop then was placed in the Frozebag, heat-sealed, and stored at −80°C for a few days. When the Frozebag containing frozen embryos was thawed, washed two times via a port using syringes (Fig 3g), and then cultured for 4 days, 50% of the embryos were recovered from the mesh bag, and none formed blastocysts. Furthermore, no embryo reached the blastocyst stage even when embryos were immediately harvested from the Frozebag after thawing and cultured on a Petri dish for 4 days (37°C, 5% $CO_2$) (Table 2). This outcome may be explained by the possibility that the Frozebag method failed to allow

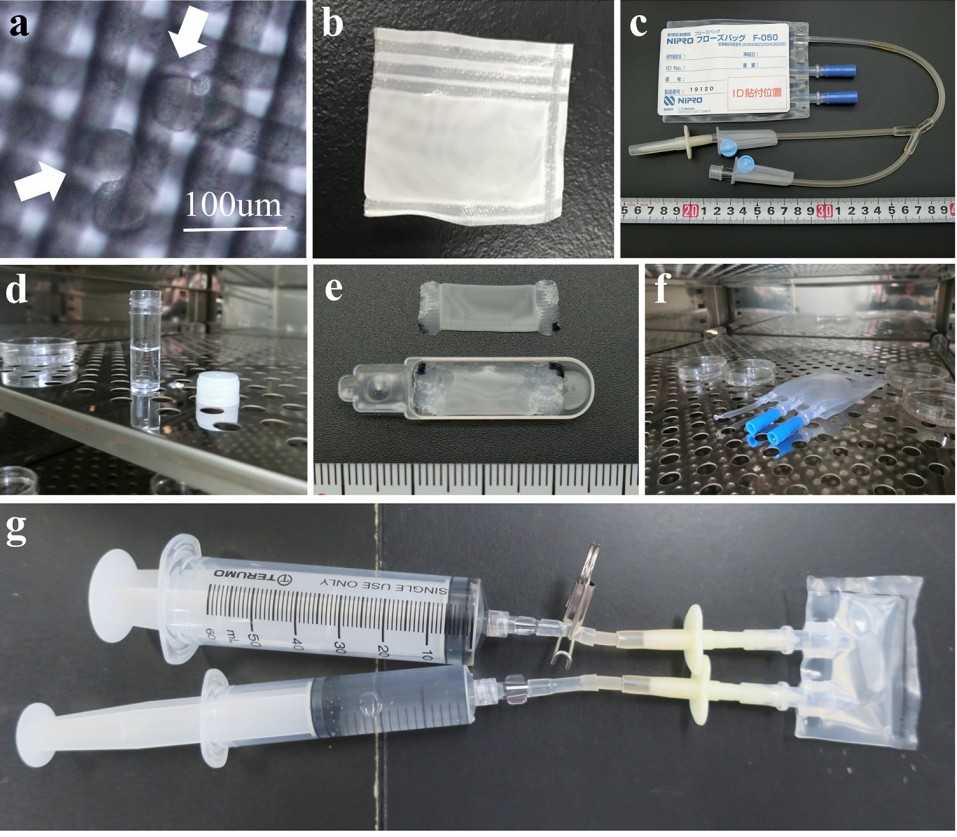

**Fig 3. High osmolarity vitrification using a Frozebag and mesh sheet.** (a) Mesh sheet and embryos (arrows). (b) Mesh bag. (c) Frozebag before modification. (d) Equilibration with CZB medium in an incubator (5% CO, 37˚C). (e) Mesh bag on the scoop. (f) Incubation of the Frozebag (5% $CO_2$, 37˚C). (g) Solution/medium exchange using 30 ml and 50-ml syringes.

exchange of the vitrification solution with the dilution solution within the restricted time limit. Furthermore, the Frozebag contained a 10-fold greater volume of EFS42.5 solution compared with the HOV method.

## Frozebag using a mesh cap

To avoid introducing a large volume of EFS42.5 solution into the Frozebag, we attempted to freeze the embryos directly on the inner surface of the Frozebag in the presence of a small

**Table 2. Recovery and blastocyst rates of embryos frozen and thawed with frozebag and mesh bag.**

| Medium | Mesh bag | Freezing | In vitro culture | No. used embryos | No. collected (%) | No. embryo developed to blastocyst (%) |
|---|---|---|---|---|---|---|
| No-equilibration | No | No | Frozebag | 20 | - | 0 |
| Equilibration | No | No | Frozebag | 48 | - | 45 (94) |
| | Yes | No | Frozebag | 50 | 49 (98) | 42 (86) |
| | Yes | Yes | Frozebag | 30 | 15 (50) | 0* |
| | Yes | Yes | Dish | 30 | 19 (63) | 0** |

Stastitial anayalsis was not perfromed due to the different experiments.

* 3 embryos reached for 4-cell.

** 1 embryo reached for poor quality morula.

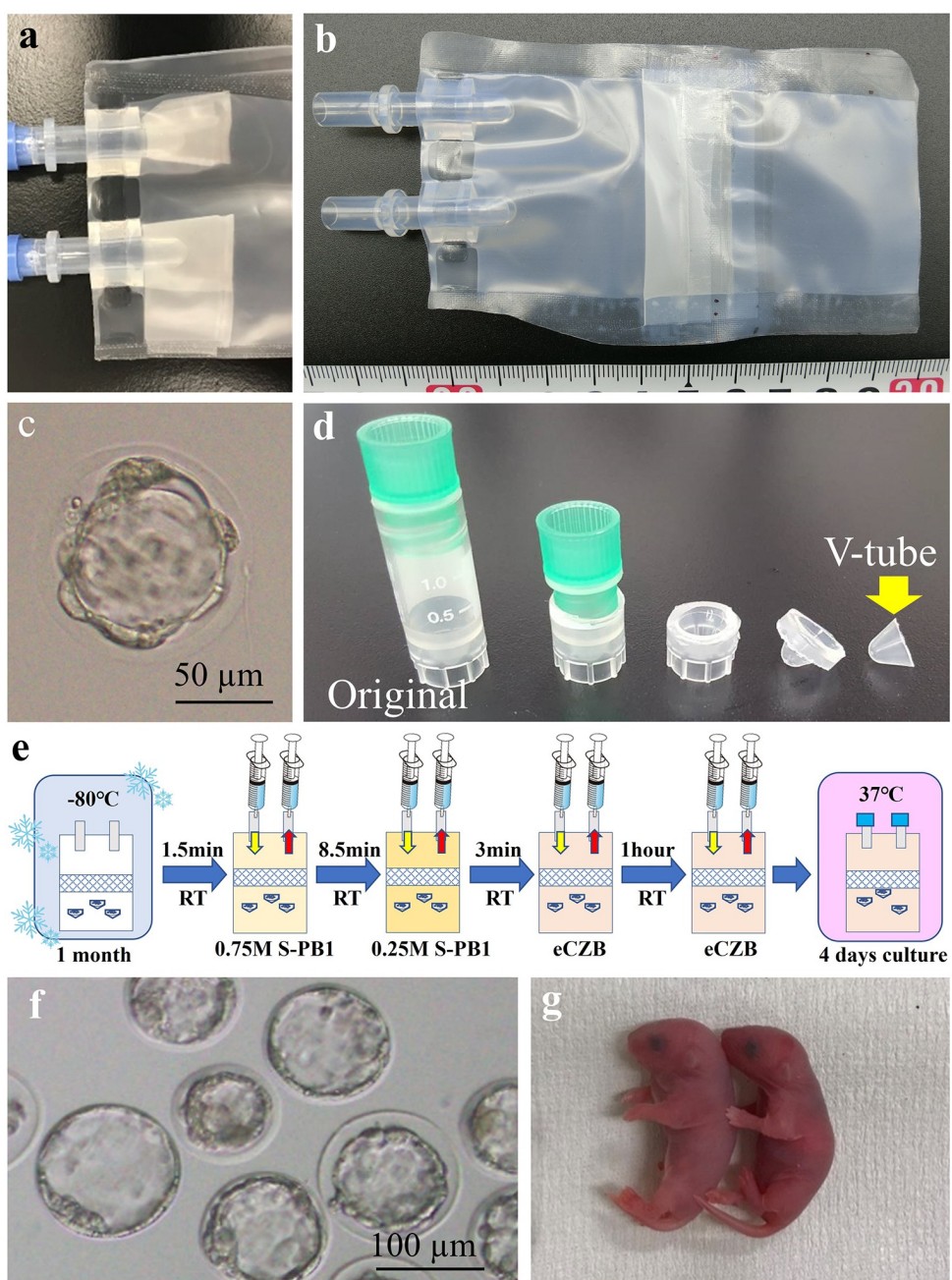

**Fig 4. Embryo Thawing and Culturing unit (ETC).** (a) Frozebag with mesh cap. (b) Frozebag with mesh wall. (c) Blastocyst collected from the Frozebag. Although the quality of this embryos was insufficient, this was the first time that an embryo was thawed, washed, and cultured for 4 days without direct contact. (d) The top of cryotube was cut to produce the smallest tube (V-tube). (e) Protocol for thawing, washing, and culturing embryos in the ETC. (f) Blastocysts were harvested from the ETC after culture for 4 days. (g) Healthy offspring developed from the morulae/blastocysts derived from 3 days cultured embryos in ETC.

amount of EFS42.5 solution. To prevent flushing out the embryos during solution exchange, the mesh sheet was converted to a cap shape (hereafter "mesh cap") and attached to the inner port of the Frozebag (Fig 4a). When fresh embryos were inserted into the Frozebag, the medium was exchanged twice. After 4 days of culture, 15 of 60 embryos were collected, among

**Table 3. Recovery and blastocyst rates of embryos frozen and thawed in ETC using a cryotube or V-tube.**

| Freezing method | Mesh | Freezing | In vitro culture | Medium change (interval) | No. used embryos | No. collected (%) | No. survived (%) | No. embryo developed to blastocyst (%) |
|---|---|---|---|---|---|---|---|---|
| **Surface** | Cap | No | Frozebag | - | 60 | 15 (25) | 15 (100) | 12 (80)* |
| | | Yes | Frozebag | 1 | 76 | 18 (24) | 0 (0) | - |
| **Cryotube** | Wall | No | Frozebag | - | 60 | 48 (80) | 43 (90)a | 39 (91)a |
| | | Yes | Frozebag | 1 | 175 | 105 (60) | 7 (7)b | 2 (29)b |
| | | Yes | Dish | 1 | 50 | 48(96) | 29 (60)c | 16 (55)b |
| **V-tube** | Wall | Yes | Frozebag | 2 (1h) | 98 | 61 (62) | 51 (84)a | 30 (59)a |
| | | Yes | Frozebag | 2 (2h) | 141 | 110 (78) | 17 (15)b | 17 (100)b |
| | | Yes | Frozebag | 3 (0.5h/2h) | 167 | 95 (57) | 19 (20)b | 19 (100)b |

Numbers with different superscripts (a-c) within the same column are significantly different (P < 0.01).

* 6 embryos were transferred into recipient and 2 offspring were obtained.

which 80% developed into blastocysts (Table 3: Surface). When embryos were frozen on the surface of the Frozebag, thawed, washed, and cultured for 4 days, 24% were recovered and none survived (Table 3).

## Frozebag equipped with a cryotube and mesh wall

We assumed that the HOV method could be performed in a cryotube [18]. The material and thickness of the Frozebag are completely different from those of the cryotube, and the difference in thermal conductivity during freezing may have damaged the embryos. Therefore, we used cryotubes to freeze the embryos and placed them in a Frozebag, which was stored at −80˚C for several days. On the other hand, the embryo recovery rate using a mesh cap from the Frozebag were very low, ranged from 24% to 25%. We after testing several other methods, we decided to use a heat sealer to attach a mesh sheet to the middle of the Frozebag to serve as a barrier (hereafter "mesh wall") to prevent flushing out the embryos (Fig 4b). When fresh embryos were inserted into cryotubes, which were placed in this modified Frozebag with mesh wall and cultured for 4 days, 80% of the embryos were recovered. Blastocytes were produced by 91% of these embryos (Table 3).

Next, embryos were frozen in cryotubes, placed in a Frozebag, and stored at −80˚C for several days. When the Frozebag was thawed, washed by exchanging the solutions, and cultured for 4 days in eCZB medium, we obtained two blastocysts from 175 used embryos (1%) or from seven survived embryos (29%). Although the morphological quality of those blastocysts was quite poor (Fig 4c), this was the first successful attempt to culture frozen embryos to produce blastocytes without directly manipulating the embryos.

The low yield of blastocysts may be explained by insufficient removal of the EFS42.5 solution from the Frozebag, even after multiple exchanges of the solution. Next, we opened the Frozebag immediately after thawing, collected the embryos, and cultured them on a Petri dish in the $CO_2$ incubator. This procedure yielded 48 of 50 embryos, among which 16 (55%) developed into blastocysts (Table 3: Cryotube). Hereafter, we called this modified Frozebag as an Embryo Thawing and Culturing unit (ETC).

## ETC using V-tube

To improve the efficiency of exchanging the EFS42.5 solution with eCZB medium, the size of the cryotubes was reduced to the extent possible (Fig 4d). We called the smallest cryotube a "V-tube." We simultaneously attempted to increase the number of medium exchanges to

completely remove residual vitrification/dilution solution from ETC (Fig 4e). Embryos were frozen using the V-tube, placed in the ETC, stored at −80°C for a few days, thawed with an additional medium exchange 1-h later, and cultured for 4 days. We recovered 61 of 98 embryos (62%), and 30 (59%) developed into good-quality blastocysts (Fig 4f, Table 3: V-tube). We also succeeded in obtaining good-quality blastocysts when one or two additional medium exchanges were performed 2-h later or 30 min and 2-h later, although the rate of survived embryo decreased to15%–20%.

To evaluate embryo quality using this system, morulae/blastocysts cultured for 3 days in ETC were transferred to mice. In a control experiment in which 2-cell stage embryos removed immediately after thawing in the ETC, were transferred to the oviduct of the recipient female, we showed a birth rate of 47%, which was comparable to birth rates of our usual frozen-thawed embryos [21]. When embryos collected after 3 days of culture in the ETC were transferred to the uterus, birth rate was 31%, which was slightly lower than the control (Table 4). However, this result seems to be within the normal range because birth rate decreases when embryos cultured *in vitro* for an extended period of time are transferred. When blastocysts were collected after 4 days of culture in the ETC and transferred to the recipient, birth rate strongly decreased. This is likely because day 4.5 stage embryos were transferred into the 2.5-dpc recipient female mice. Importantly, this outcome demonstrates that the ETC generated normal morulae/blastocysts.

## Preservation period of embryos at −80°C and reproducibility of this method

When the µG experiment is conducted on the ISS, storage in the −95°C freezer may exceed 1 month. Therefore, we determined our ability to preserve frozen embryos at −80°C in the ETC. When the ETC was stored for 1 month at −80°C, 17 of 52 (32.7%) embryos developed into blastocysts. The blastocyst generation rate decreased with longer storage, although some embryos developed into blastocysts after storage for 3 months (Table 5). However, the HOV method we used for embryo freezing allows frozen embryos to be stored at −80°C for at least 1 month and we believe it is feasible to conduct experiments on the ISS within a month of

**Table 4. Production of offspring from embryos thawed and cultured in the ETC.**

| Culture period in ETC (days) | Stage of transferred embryos | No. transferred embryos | No. implanted (%) | No. offspring (%) |
|---|---|---|---|---|
| 0 | 2-cell | 17 | 2 (12) | 8 (47) |
| 3 | Morula/Blastocyst | 68 | 32 (47) | 21 (31) |
| 4 | Blastocyst | 16 | 8 (50) | 2 (13) |

There were no significant difference between all group (P>0.01).

**Table 5. Development of blastocysts obtained from embryos cryopreserved at −80°C for up to three months and performed by inexperienced people.**

| Preservation period at -80 °C | No. used embryos | No. collected (%) | No. embryo developed to blastocyst (%) |
|---|---|---|---|
| 1 M | 150 | 52 (35) | 17 (33)a |
| 1 M* | 180 | 89 (49) | 18 (20)a |
| 2 M | 246 | 170 (69) | 13 (8)b |
| 3 M | 231 | 145 (63) | 4 (3)b |

Numbers with different superscripts (a, b) within the same column are significantly different (P < 0.01).

* This experiment were performed by inexperienced people.

arrival. Furthermore, we determined whether inexperienced people were able to obtain blastocysts using the ETC. For this purpose we used 2 ETCs with 180 vitrified embryos. Two inexperienced people thawed and cultured the embryos. Although the rate of blastocyst generation was not high, both person obtained good blastocysts (Table 5).

## Discussion

Through this research, we aim to develop a new device that can be used for conducting experiments on the ISS and determine whether gravity is necessary for mammalian embryo development. Although developing a transparent culture device that would allow observation of embryos during experiment on the ISS could not be achieved, we have developed a device—ETC—that can thaw frozen embryos on the ISS and develop them into good-quality blastocysts.

Space experiments require embryos to be stored at −95°C; thus, the HOV method was adopted in this study, which allows cryopreservation of embryos at −80°C for at least one month [18]. However, the HOV method contains a high concentration of cryoprotectants with high osmolality, which may cause some damage to the embryos [22, 23]. Healthy offspring was obtained from cryopreserved embryos using either the HOV method or ETC. Therefore, the quality of embryos does not hinder the application of experiments on Earth, but the development may be negatively affected when embryos were cultured in severe space experiments. Recently, Qiu et al. reported that embryos could be cryopreserved with low concentration of cryoprotectants at −80°C for one month [24, 25]. Using this method, medium exchange using a hollow fiber tube, which was used in this early study, may be feasible. If so, not only improve the embryo recovery rates from ETC, but also would allow observation of the embryo during space experiments.

Although the ETC could achieve a high developmental rate for thawed two-cell embryos, we cannot recover all embryos after exchanging several solutions and medium. Given the high cost of space experiments, we aimed to achieve a 100% recovery rate of embryos from ETCs, but the launch schedule of the rocket did not allow the scope for further improvement. Conducting experiments on the ISS will simultaneously examine blastocyst development between microgravity and artificial-1G environments. Therefore, the experiments will be successful even if only a few embryos are recovered in both groups. In addition, this aim of this study was to develop a device that will allow untrained astronauts to reliably thaw and culture embryos, rather than obtain a perfect recovery rate of embryos.

In September 6, 2021, an experiment using this device was conducted on the ISS, proving the usefulness of the device. However, the embryo recovery rate obtained by the ETC was lower than that shown in this study (submitted). We tried various improvements right up to launch, such as a good equilibration medium for embryo culture in ETC [20], and identified the most appropriate stage of mouse two-cell embryos for the present project [21]. However, space experiments can be challenging. For example, no convection of liquid is available on the ISS, which may hinder complete removal of cryoprotectants from ETC. Therefore, further improvement of ETC is necessary to conduct difficult experiments on the ISS, such as *in vitro* fertilization.

The development of this method does not require skillful handling of embryos. If the recovery rate of embryos reaches 100%, this device could have many other uses, apart from space experiments. Transporting animals is time consuming and labor intensive given the quarantine, but transporting frozen embryos in ETCs with dry ice makes it easier. Fertility clinics require training for embryologists, but this device could shorten the training period. Therefore, we must determine where the embryos are flushed out from ETC and improve the ETC

to achieve a 100% embryo recovery rate. In addition, determining whether ETCs can be used in other animal species other than mice and can be stored for longer periods at −80˚C is necessary.

## Methods

### Mice

Female and male BDF1 mice were obtained from SLC Inc. (Hamamatsu, Japan). ICR mice were bred in our mouse facility. Embryos were collected from 8–10 weeks female mice. Surrogate pseudo-pregnant ICR females, which were used as embryo recipients, were mated with vasectomized ICR males whose sterility was demonstrated previously. On the day of the experiment, or upon completion of all experiments, the mice were killed using $CO_2$ inhalation or cervical dislocation. All animal experiments were conducted in accordance with the Guide for the Care and Use of Laboratory Animals and were approved by the Institutional Committee of the Laboratory of Animal Experimentation of the University of Yamanashi (reference number: A29-24), which followed the ARRIVE guidelines.

### Media

HEPES-CZB (H-CZB) medium [26] was used to collect embryos from oviducts, and CZB medium [27] was used to culture embryos in a humidified incubator (5% $CO_2$, 37˚C). Vitrification solution for HFT method: Equilibration and vitrification solutions were used for vitrification, and the rewarming, dilution, and washing solution were used for thawing [19]. The vitrification solutions for the HOV method included EFS20 and EFS42.5. The embryos were thawed using 0.25 M and 0.75 M sucrose-PB1 (S-PB1) [18]. Equilibrated CZB (eCZB) medium was prepared as previously described [20]. Briefly, CZB medium was used to fill the container, the lid was placed slightly off-center, and the container was then placed in the $CO_2$ incubator for >24 h to equilibrate the $CO_2$ concentration of the medium with that of the atmosphere. The eCZB medium was prepared 1 day before commencing the experiment.

### *In vivo* fertilization

Female mice were induced to superovulate by injection of 5 international units (IU) of equine chorionic gonadotropin (eCG) followed 48-h later by injection of 5 IU of human chorionic gonadotropin (hCG). These female mice were immediately mated and examined the following day for vaginal plugs, after which they were separated from the males. The next day, 2-cell embryos were recovered by flushing the oviducts with H-CZB medium and incubating then in CZB medium at 37˚C in an atmosphere containing 5% $CO_2$.

### Preparation of HFTs

HFTs are straw-like structures with a reticulated membrane that tolerate freezing at −196˚C and are nontoxic for mammalian embryos. HFTs are used to simplify the vitrification of embryos [19]. We reasoned that if embryos were frozen inside an HFT, the astronauts could thaw and culture the embryos without directly contacting them. The internal diameter of an HFT (185 μm) and the pore sizes of the membrane range between 5–15 nm, which allows exchange solutions through HFT membrane but not lose any embryos (Fig 2a). The HFT comprises a transparent film, which allows visual observations of embryos morphology during the 4-day culture.

### HFT vitrification

The vitrification of 2-cell embryos using the HFT was performed using a modified published method [19]. Briefly, approximately 10 2-cell embryos were placed in the equilibration solution and then aspirated into the HFT (Fig 2a), which was sealed by pinching its ends using a tweezers. Following equilibration (5–7 min), the HFT containing embryos was transferred to the vitrification solution placed on an ice bath. Next, the embryos were aspirated into a 1-ml syringe attached with a small pipette tip (for 20-μl use) (Fig 2b and 2c), and the syringe was immersed in $LN_2$ within 1–2 min (Fig 2d).

### OptiCell method

OptiCell (Thermo Fisher Scientific) is a commercially available cell culture device, which comprises a transparent $CO_2$-permeable membrane and two ports for medium exchange. For thawing, the syringe was removed from the $LN_2$ (Fig 2e) and then the pipette tip attached to the syringe was inserted into a port of the OptiCell, previously filled with thawing solution. Within 1 min, the HFT with embryos was ejected from the syringe into the warm OptiCell (Fig 2f). Next, a 10-ml syringe containing dilution solution and an empty 10-ml syringe were attached to both ports of the OptiCell. Within 2 min, dilution solution was injected into the OptiCell, and the thawing solution was removed using the empty syringe (Fig 2g). After 3 min, the dilution solution was similarly replaced with washing solution, after 5 min, the washing solution was replaced with CZB medium, and then the OptiCell with the HFT was placed in a $CO_2$ incubator for 4 days.

### Mini-plate method

Mini-plates comprise a transparent $CO_2$-permeable membrane and two ports as in the Opti-Cell, but are much smaller (3 cm × 2 cm) than those of the OptiCell (Fig 2h). Experiments using a Mini-plate were conducted under the same conditions used for the OptiCell, with the exception that a 21-gage needle was used to insert the HFT into the Mini-plate, because the inner diameter of port was too small and tight. In other experiments, the vitrification solution was added to the Mini-plate, and the equilibrated embryos in the HTF were the injected into the Mini-plate, which was placed in the vitrification solution. Within 1 min, the Mini-plate was immersed in $LN_2$.

### Cell Experimental Unit (CEU) method

Embryos in the HFT were prepared using the OptiCell method, except that the HFT with frozen embryos was placed on the center of the window of the disassembled CEU, assembled within time limit, and stored in $LN_2$. In some experiments, vitrification solution was placed on the windows of the disassembled CEU, the equilibrated embryos in the HTF were placed into the vitrification solution, and then the CEU was assembled and immersed in $LN_2$.

### New Cell Experimental Unit (NCEU) method

To serve as a control, the HFT with fresh embryos was pinched onto the center of the NCEU using a silicone flipper (Fig 2n), and a gas-permeable membrane, which was used to cover the culturing area of the NCEU, was fixed using a nontoxic adhesive. After medium exchange, the NCEU was placed in an incubator (5% $CO_2$, 37˚C) for 4 days.

## Frozebag with a mesh bag

The Frozebag (NIPRO Co. Japan), which provides two ports for medium exchange, is easily processed using a heat sealer, withstands $LN_2$, but is not gas-permeable (Fig 3c). The pore size of the mesh sheet (NBC meshtec Inc, Japan) is approximately 20 μm (Fig 3b), and the embryos cannot be observed during the experiment. However, in the absence of an alternative, we abandoned to observe embryos on the ISS. We used a heat sealer to convert the sheet into a 1 cm × 2-cm bag (hereafter "mesh bag") (Fig 3b).

To serve as a suitable control experiment, the mesh bag was placed on the scoop, which was used as a saucer (NIPRO Cell Sleeper, Japan), and filled with eCZB medium (Fig 3e). Next, 10–20 fresh embryos were inserted into the mesh bag through a small slit. The scoop with the mesh bag was placed in a Frozebag, which was heat-sealed and filled with eCZB medium through the port (Fig 3g). The Frozebag was then placed in an incubator (5% $CO_2$, 37˚C) for 4 days. For experiment, embryos were transferred to the vitrification solution (EFS20) on the Petri dish and incubated for 2 min at room temperature. Next, the embryos were transferred into vitrification solution (EFS42.5) on the Petri dish, and within 1 min, were transferred into the mesh bag, which was filled with EFS42.5 on the scoop. The scoop was then immersed in $LN_2$. After vitrification, the scoop was inserted into the Frozebag in $LN_2$, which was then removed and immediately heat-sealed, after which it was returned to $LN_2$ and stored at −80 ºC.

## Thawing and culturing embryos using a Frozebag

For thawing embryos, the Frozebag was removed from the freezer and injected with the first dilution solution (0.75 M S-PB1) through a 30-ml syringe via the port within 1.5 min. After 8.5 min, this solution was exchanged with the second dilution solution (0.25 M S-PB1) and the waste solution was simultaneously drained from another port. After 3 min, this solution was exchanged with the culture medium (eCZB). Finally, the medium was similarly exchanged with eCZB medium at one hour later and placed in an incubator (5% $CO_2$, 37˚C) for 4 days.

## Collection of embryos from the Frozebag

Upon completion of the culture period, the cut the end of the Frozebag and the mesh bag with medium were transferred to a 10-cm dish. The mesh bag was opened using tweezers and gently shaken to release the embryos. The embryos were observed using a microscope to determine the recovery and developmental rates. In some experiments, the embryos were further cultured on a dish in an incubator (5% $CO_2$, 37˚C).

## Frozebag with a mesh cap

To prevent the loss of embryos during solution exchange, a mesh sheet cap was attached to the inner port of the Frozebag (Fig 4a). To serve as a control, fresh embryos were placed on the surface of a Frozebag, filled with eCZB medium, and placed in an incubator (5% $CO_2$, 37˚C) for 4 days. Embryos were transferred into a 50 μl-drop of vitrification solution (EFS 20) on the Perti dish for 2 min, transferred into a 50 μl-drop of vitrification solution (FEF42.5) on the inside surface of the Frozebag, which was heat-sealed within 1 min. Next, the Frozebag was immediately frozen in $LN_2$ and stored at −80˚C. The embryos were thawed as described above.

## Frozebag with a cryotube or a V-tube

To prevent the loss of embryos during solution exchange, a mesh sheet wall was attached to the middle of the Frozebag (Fig 4b). Since the Frozebag and mesh sheet do not stick together

by heat-sealed, a piece of plastic bag (Ziploc, Asahi KASEI, Japan) was placed in the gap between Frozebag and mesh sheet, which was used as an adhesive to heat seal the bags, making them adhere perfectly. For vitrification, 10–30 embryos were suspended in EFS20 equilibrium solution for 2 min and then transferred into a cryotube (Sumitomo Bakelite, Japan) or V-tube, fabricated in-house, (Fig 4d) containing 50 µL of EFS42.5. After 1 min, the cryotube or V-tube was plunged directly into $LN_2$. These tubes were then placed in the Frozebag under $LN_2$, heat-sealed as described above, and then stored at −80 ºC. Thawing was performed as described above.

## Embryo transfer

Embryos (2-cell stage) collected from the Frozebag just after thawing, morulae/blastocysts cultured for 3 days, or blastocysts cultured for 4 days were transferred to the oviduct of day 0.5 or uteri of day 2.5 pseudopregnant mice mated with vasectomized males [28]. Embryos (n = 5–8) were transferred into each oviduct. On day 18.5, the offspring were delivered by cesarean section and allowed to mature.

## Statistical analysis

Survival, developmental, and birth rates were evaluated using chi-squared tests. Statistically significant differences between the variables are defined as $p < 0.01$. In some cases, Chi-squared tests may not be appropriate given the low number of embryos. However, this study aimed to develop a new device, and the results of the main experiments were either a success or failure. Therefore, statistical analysis was not helpful for the next device development.

## Acknowledgments

We thank Dr. A. Higashibata, Dr. T. Yamamori, Dr T. Kohda, Dr S. Kishigami, Dr M. Ooga, Dr Y. Fujimoto, Mrs. I. Osada, C. Yamaguchi, and Y. Kanda for assistance in preparing this manuscript. The authors would like to thank Enago for the English language review.

## Author Contributions

**Conceptualization:** Sayaka Wakayama, Teruhiko Wakayama.

**Formal analysis:** Sayaka Wakayama, Teruhiko Wakayama.

**Funding acquisition:** Teruhiko Wakayama.

**Investigation:** Sayaka Wakayama, Mariko Soejima, Yasuyuki Kikuchi, Erika Hayashi, Natsuki Ushigome, Ayumi Hasegawa, Keiji Mochida, Tomomi Suzuki, Chiaki Yamazaki, Toru Shimazu, Hiromi Sano, Masumi Umehara, Hitomi Matsunari, Atsuo Ogura, Hiroshi Nagashima, Teruhiko Wakayama.

**Methodology:** Sayaka Wakayama.

**Project administration:** Teruhiko Wakayama.

**Supervision:** Teruhiko Wakayama.

**Validation:** Sayaka Wakayama, Teruhiko Wakayama.

**Writing – original draft:** Sayaka Wakayama, Teruhiko Wakayama.

**Writing – review & editing:** Sayaka Wakayama, Atsuo Ogura, Hiroshi Nagashima, Teruhiko Wakayama.

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
