## [Decision Letter · Decision Letter 0]

9 Aug 2022

PONE-D-22-17378Development of a new device for manipulating frozen mouse 2-cell embryos on the International Space StationPLOS ONE

Dear Dr. Wakayama,

Thank you for submitting your manuscript to PLOS ONE. After careful consideration, we feel that it has merit but does not fully meet PLOS ONE’s publication criteria as it currently stands. Therefore, we invite you to submit a revised version of the manuscript that addresses the points raised during the review process.

Dear authors, please take the time to carefully review the issues raised by reviewer #2.

We look forward to receiving your revised manuscript.

Kind regards,

Marcelo Fábio Gouveia Nogueira, Associate Professor, Ph. D.

Academic Editor

PLOS ONE

Journal Requirements:

"This work was partially funded by the Naito Foundation (S.W) and Takahashi Industrial and Economic Research Foundation (189) to S.W.; Asada Science Foundation (T.W) and the Canon Foundation (M20-0006) to T.W. The authors would like to thank Enago for the English language review."

"Additional Information

Declaration of Interests

Declare that there is no conflict of interest that could be perceived as prejudicing the impartiality of the research reported."

Reviewers' comments:

Reviewer's Responses to Questions

**Comments to the Author**

1. Is the manuscript technically sound, and do the data support the conclusions?

Reviewer #1: Yes

Reviewer #2: Yes

2. Has the statistical analysis been performed appropriately and rigorously? 

Reviewer #1: Yes

Reviewer #2: No

3. Have the authors made all data underlying the findings in their manuscript fully available?

Reviewer #1: Yes

Reviewer #2: Yes

4. Is the manuscript presented in an intelligible fashion and written in standard English?

Reviewer #1: Yes

Reviewer #2: Yes

5. Review Comments to the Author

Reviewer #1: Authors developented new device which helps to thaws, wash and culture frozen 2 cell embryos. The advantages of the device is that embryos may be manipulated by inexperienced people without direct contact.

Even if the preliminary results are promising, optimizing the recovery and development rate is necessary for full succes.

Reviewer #2: The authors developed a simple system for thawing and culturing embryos, which in theory would allow untrained personnel such as astronauts to perform embryo experiments on Earth and on the International Spatial Station (ISS). Using a mice model, the authors described the development and validation of the ETC, which thaws, washes, and cultures frozen mouse 2-cell embryos without direct contact. Despite, the ETC achieves a high developmental rate of thawed 2-cell embryos, the recovery rate was not optimal after exchanging several solutions. In agreement with authors, the technique must be better developed before doing it in the ISS. My major concern is regarding to the statistical analyses. The embryo was the experimental unit, however, to perform a Chi-square test, the presented number if not enough. Finally, the present study is a well written, the statistics was done appropriately with good conclusions. Therefore, I do recommend the acceptance for publication in Plos One after major revisions.

Specific Comment:

Abstract

The abstract is well written. However, as suggestion, please briefly present the major results and explain the statistical analysis.

Keywords: Use different words previously used in the text. The keywords are used to better expose your work.

Introduction:

This section needs better review on the latest vitrification achievements.

Material and methods

This section is well written. However, there is missing an experimental design The ideas are not easy to follow, and it may compromise the reading. I strongly suggest adding an Experimental Design explanation, introducing the different experiments that will be presented.

The statistical analyses is poor. The Chi-square test have not enough power for some of the present experiments, based on the low number of used embryos. As suggestion, continue the experiments, increasing the number of replicates.

Results

Tables: Do not use solid lines between groups.

Tables: Present better the tables titles. It may be self-explanatory, in the present form, they aren’t.

Discussion

This section needs to be improved. Is too simple the discussion. Some considerations are important and must be addressed such the main vitrification techniques; result for other species; novelty of the hypothesis. Besides, after reading the discussion, only two references were cited.

6. PLOS authors have the option to publish the peer review history of their article (what does this mean?). If published, this will include your full peer review and any attached files.

Reviewer #1: No

Reviewer #2: **Yes: **José Felipe Warmling Sprícigo

---

## [Author Response · Author response to Decision Letter 0]

28 Aug 2022

Response to Reviewers (please find attached file too)

5. Review Comments to the Author

Reviewer #1: Authors developented new device which helps to thaws, wash and culture frozen 2 cell embryos. The advantages of the device is that embryos may be manipulated by inexperienced people without direct contact.

Even if the preliminary results are promising, optimizing the recovery and development rate is necessary for full succes.

We had been developing this device since 2012 and continued to improve the recovery and developmental rate of embryo, right up to launch. However, the rocket launch schedule was strictly set and cannot be extended further, to avoid cancellation. Although full success of the device was not achieved, here we could demonstrate that inexperienced people could be able to successfully obtain a blastocyst using this device. This suggest that the minimum success of this study had been achieved, and therefore, we conducted a space experiment. In that experiment, embryos were cultured in microgravity or artificial-1G environments created simultaneously on the ISS. Therefore, the experiment will be successful even if only a few embryos are recovered from ETCs in both groups. 

This paper aims to describe the device used in the space experiment, and will be cited in a paper on the space experiments that we will be submitting soon. Although we would like to continue to improve the device, we must present it in the form it was used in the space experiment. We hope for your consideration.

Reviewer #2: The authors developed a simple system for thawing and culturing embryos, which in theory would allow untrained personnel such as astronauts to perform embryo experiments on Earth and on the International Spatial Station (ISS). Using a mice model, the authors described the development and validation of the ETC, which thaws, washes, and cultures frozen mouse 2-cell embryos without direct contact. Despite, the ETC achieves a high developmental rate of thawed 2-cell embryos, the recovery rate was not optimal after exchanging several solutions. In agreement with authors, the technique must be better developed before doing it in the ISS. My major concern is regarding to the statistical analyses. The embryo was the experimental unit, however, to perform a Chi-square test, the presented number if not enough. Finally, the present study is a well written, the statistics was done appropriately with good conclusions. Therefore, I do recommend the acceptance for publication in Plos One after major revisions.

Thank you for the valuable feedback. We have rewritten the manuscript as per your comments, particularly the Discussion section.

Specific Comment:

Abstract

The abstract is well written. However, as suggestion, please briefly present the major results and explain the statistical analysis.

We thank you for this advice. We have added the following text to the Abstract section.

“Whether mammalian embryos develop normally under microgravity is yet to be determined. However, embryos are too small to be handled by inexperienced astronauts who orbit Earth on the International Space Station (ISS). Here, we describe the development of a new device that allows astronauts to thaw and culture frozen mouse two-cell embryos on the ISS without any practice. First, we developed several new devices using a hollow fiber tube that allows thawing of the embryo without direct contact and observing embryonic development. The recovery rate of embryos was over 90%, and the developmental rate of the blastocyst was over 80%. However, general vitrification requires liquid nitrogen, which is not available on the ISS. Therefore, we developed another new device—embryo thawing and culturing unit (ETC)—using high-osmolarity vitrification, which could preserve frozen embryos at −80°C for several months. Embryos flushed out of the ETC during thawing and washing were protected using a mesh sheet. Although the recovery rate of embryos after thawing were not high (24%–78%) and embryonic development in ETC could not be observed, thawed embryos formed blastocysts after four days of culture (29%–100%) without direct contact. Thus, the ETC could be used by untrained astronauts to thaw and culture frozen embryos on the ISS. In addition, this device will be an important advancement in fields such as clinical infertility and animal biotechnology, when the recovery rate of embryos was improved by nearly 100%.”

Keywords: Use different words previously used in the text. The keywords are used to better expose your work.

We have modified the keywords and added “Development” and “Microgravity.”

Introduction:

This section needs better review on the latest vitrification achievements.

We have edited this section as per your suggestion and included the following sentences.

Line 82-

“The abovementioned experiments were simulations performed on the ground. We aimed to determine whether the same result would be obtained in the �G of real space, such as the ISS. Frozen embryos must be launched, thawed, and cultured on the ISS to study the effects of �G on embryonic development, because mouse embryos can be cultured in vitro for only 4 to 5 days. However, two factors hindered the application of such experiments on the ISS. First, handling of embryos by untrained personnel not only in orbit but also on the ground is not feasible given their small size (80–100 �m). If the cells are sticking to the bottom of the culture device, they will not flow out when culture medium was exchanged using a pump or syringe by astronauts on the ISS [15]. However, considering that the embryos are floating in the medium, they can easily flow out when medium was changed. Therefore, developing a new device that prevents embryos from flowing out when astronauts perform space experiment is necessary.

Another factor hindering the application of space experiments on the ISS is the cryopreservation temperature of frozen embryos. Frozen embryos should be maintained below −130°C during cryopreservation to avoid cryodamage [16, 17]. If embryos are exposed to high temperature, intracellular ice crystals form, thereby irreversibly damaging the embryos. However, liquid nitrogen (LN2) is not available on the ISS, and the temperature of the coldest freezer is −95°C. Furthermore, the frozen embryos must be preserved for a few weeks or months on the ISS before the commencement of experiment, because space experiments cannot be performed immediately after arrival at the ISS. During the study, Mochida et al. developed high-osmolality vitrification (HOV), which enable the storage of the embryos at −80°C for several months [18].

In this study, we developed a simple system for thawing and culturing embryos to address the aforementioned challenges, which will allow us to preserve embryos at −80°C for a few months and untrained personnel, such as astronauts, to perform embryo experiments on Earth and on the ISS.”

Material and methods

This section is well written. However, there is missing an experimental design The ideas are not easy to follow, and it may compromise the reading. I strongly suggest adding an Experimental Design explanation, introducing the different experiments that will be presented.

We added a new figure 1, as per your suggestion.

Fig. 1. Study outline. Diagram showing the performance and weaknesses of the device, completed devices, and blastocysts

The statistical analyses is poor. The Chi-square test have not enough power for some of the present experiments, based on the low number of used embryos. As suggestion, continue the experiments, increasing the number of replicates.

We had been developing this device since 2012 and continued to improve the recovery and developmental rate of embryo, right up to launch. However, the rocket launch schedule was strictly set and cannot be extended further, to avoid cancellation. Although full success of the device was not achieved, here we could demonstrate that inexperienced people could be able to successfully obtain a blastocyst using this device. This suggest that the minimum success of this study had been achieved, and therefore, we conducted a space experiment. In that experiment, embryos were cultured in microgravity or artificial-1G environments created simultaneously on the ISS. Therefore, the experiment will be successful even if only a few embryos are recovered from ETCs in both groups. 

This paper aims to describe the device used in the space experiment, and will be cited in a paper on the space experiments that we will be submitting soon. Although we would like to continue to improve the device, we must present it in the form it was used in the space experiment. We hope for your consideration.

We have added the following sentences to the Method section:

Line 507-

“In some cases, Chi-squared tests may not be appropriate given the low number of embryos. However, this study aimed to develop a new device, and the results of the main experiments were either a success or failure. Therefore, statistical analysis was not helpful for the next device development” 

Results

Tables: Do not use solid lines between groups.

We have removed these lines from the tables.

Tables: Present better the tables titles. It may be self-explanatory, in the present form, they aren’t.

We have modified table titles as follows:

New title

Table 1. Recovery and blastocyst rates of embryos frozen and thawed in new devices using a hollow fiber tube

Table 2. Recovery and blastocyst rates of embryos frozen and thawed with frozebag and mesh bag

Table 3. Recovery and blastocyst rates of embryos frozen and thawed in ETC using a cryotube or V-tube

Table 5. Development of blastocysts obtained from embryos cryopreserved at −80°C for up to three months and performed by inexperienced people

Discussion

This section needs to be improved. Is too simple the discussion. Some considerations are important and must be addressed such the main vitrification techniques; result for other species; novelty of the hypothesis. Besides, after reading the discussion, only two references were cited.

We thank you for the advice. We have completely rewritten this section as follows. In the text (line 326), we wrote “submitted,” but we will submit this manuscript within a few weeks.

 “Discussion

Through this research, we aim to develop a new device that can be used for conducting experiments on the ISS and determine whether gravity is necessary for mammalian embryo development. Although developing a transparent culture device that would allow observation of embryos during experiment on the ISS could not be achieved, we have developed a device—ETC—that can thaw frozen embryos on the ISS and develop them into good-quality blastocysts.

Space experiments require embryos to be stored at −95°C; thus, the HOV method was adopted in this study, which allows cryopreservation of embryos at −80°C for at least one month [18]. However, the HOV method contains a high concentration of cryoprotectants with high osmolality, which may cause some damage to the embryos [22, 23]. Healthy offspring was obtained from cryopreserved embryos using either the HOV method or ETC. Therefore, the quality of embryos does not hinder the application of experiments on Earth, but the development may be negatively affected when embryos were cultured in severe space experiments. Recently, Qiu et al. reported that embryos could be cryopreserved with low concentration of cryoprotectants at −80°C for one month [24, 25]. Using this method, medium exchange using a hollow fiber tube, which was used in this early study, may be feasible. If so, not only improve the embryo recovery rates from ETC, but also would allow observation of the embryo during space experiments.

Although the ETC could achieve a high developmental rate for thawed two-cell embryos, we cannot recover all embryos after exchanging several solutions and medium. Given the high cost of space experiments, we aimed to achieve a 100% recovery rate of embryos from ETCs, but the launch schedule of the rocket did not allow the scope for further improvement. Conducting experiments on the ISS will simultaneously examine blastocyst development between microgravity and artificial-1G environments. Therefore, the experiments will be successful even if only a few embryos are recovered in both groups. In addition, this aim of this study was to develop a device that will allow untrained astronauts to reliably thaw and culture embryos, rather than obtain a perfect recovery rate of embryos.

In September 6, 2021, an experiment using this device was conducted on the ISS, proving the usefulness of the device. However, the embryo recovery rate obtained by the ETC was lower than that shown in this study (submitted). We tried various improvements right up to launch, such as a good equilibration medium for embryo culture in ETC [20], and identified the most appropriate stage of mouse two-cell embryos for the present project [21]. However, space experiments can be challenging. For example, no convection of liquid is available on the ISS, which may hinder complete removal of cryoprotectants from ETC. Therefore, further improvement of ETC is necessary to conduct difficult experiments on the ISS, such as in vitro fertilization.

The development of this method does not require skillful handling of embryos. If the recovery rate of embryos reaches 100%, this device could have many other uses, apart from space experiments. Transporting animals is time consuming and labor intensive given the quarantine, but transporting frozen embryos in ETCs with dry ice makes it easier. Fertility clinics require training for embryologists, but this device could shorten the training period. Therefore, we must determine where the embryos are flushed out from ETC and improve the ETC to achieve a 100% embryo recovery rate. In addition, determining whether ETCs can be used in other animal species other than mice and can be stored for longer periods at −80°C is necessary.

---

## [Decision Letter · Decision Letter 1]

21 Sep 2022

Development of a new device for manipulating frozen mouse 2-cell embryos on the International Space Station

PONE-D-22-17378R1

Dear Dr. Wakayama,

We’re pleased to inform you that your manuscript has been judged scientifically suitable for publication and will be formally accepted for publication once it meets all outstanding technical requirements.

Kind regards,

Marcelo Fábio Gouveia Nogueira, Associate Professor, Ph. D.

Academic Editor

PLOS ONE

Additional Editor Comments (optional):

Reviewers' comments:

Reviewer's Responses to Questions

**Comments to the Author**

1. If the authors have adequately addressed your comments raised in a previous round of review and you feel that this manuscript is now acceptable for publication, you may indicate that here to bypass the “Comments to the Author” section, enter your conflict of interest statement in the “Confidential to Editor” section, and submit your "Accept" recommendation.

Reviewer #1: All comments have been addressed

Reviewer #2: All comments have been addressed

2. Is the manuscript technically sound, and do the data support the conclusions?

Reviewer #1: Yes

Reviewer #2: Yes

3. Has the statistical analysis been performed appropriately and rigorously? 

Reviewer #1: Yes

Reviewer #2: Yes

4. Have the authors made all data underlying the findings in their manuscript fully available?

Reviewer #1: Yes

Reviewer #2: Yes

5. Is the manuscript presented in an intelligible fashion and written in standard English?

Reviewer #1: Yes

Reviewer #2: Yes

6. Review Comments to the Author

Reviewer #1: (No Response)

Reviewer #2: The authors were careful to review the suggestions. As a last suggestion, focus on explaining the statistical analysis more objectively.

7. PLOS authors have the option to publish the peer review history of their article (what does this mean?). If published, this will include your full peer review and any attached files.

Reviewer #1: No

Reviewer #2: **Yes: **José Felipe Warmling Spricigo
